# The Effects of Subjective Wellbeing and Self-Rated Health on Lifetime Risk of Cardiovascular Conditions in Women

**DOI:** 10.3390/ijerph20146380

**Published:** 2023-07-17

**Authors:** Erum Z. Whyne, Jihun Woo, Haekyung Jeon-Slaughter

**Affiliations:** 1VA North Texas Health Care System, Dallas, TX 75216, USA; erum.whyne@gmail.com; 2Department of Kinesiology & Health Education, The University of Texas at Austin, Austin, TX 78712, USA; 3Department of Internal Medicine, University of Texas Southwestern Medical Center, Dallas, TX 75319, USA

**Keywords:** women’s heart disease, cardiovascular disease, subjective wellbeing, life satisfaction, self-rated health

## Abstract

Subjective wellbeing may predict future health conditions, and lower self-rated physical health (SRH) is associated with the presence of chronic conditions, such as cardiovascular disease (CVD). This study examines whether subjective wellbeing and SRH predict long-term CVD conditions for women using the Midlife in the United States study. The study cohort includes 1716 women participants who completed waves 1 (1995–1996), 2 (2004–2006), and 3 (2013–2014). Data on demographics, chronic conditions of diabetes and CVD, subjective wellbeing (life satisfaction, positive affect, and negative affect), and SRH were collected repeatedly at each wave. Multiple logistic regressions were conducted to test whether subjective wellbeing was associated with a lifetime CVD risk. Greater life satisfaction was significantly associated with a lower risk of CVD at 10 years (odds ratio (OR): 0.83; 95% confidence interval (CI): 0.74–0.95) and 19 years (OR: 0.83; 95% CI: 0.74–0.93), while positive and negative affects were not significantly associated. Additionally, better physical SRH significantly lowered odds of having cardiovascular conditions at both 10 years (OR: 0.79; 95% CI 0.68–0.92) and 19 years (OR 0.74; 95% CI: 0.64–0.86). Measures of life satisfaction and SRH can be used as additional CVD screening tools.

## 1. Introduction

Cardiovascular disease (CVD) is the leading cause of mortality for women, accounting for 35% of deaths in 2019 [1], and costs the United States (US) approximately 229 billion USD per year [2]. Despite the growing awareness of sex disparities in CVD, it is still understudied and not well understood [3,4,5]. Sex-difference analysis in large epidemiology studies is a commonly accepted analysis, which often leads to underestimation of importance of female sex in CVD risk due to an under-representation of women and a male-dominant CVD risk model [6]. The female subgroup analysis is warranted to elucidate female sex-specific CVD risk factors. While traditional CVD risk factors—hypertension, hyperlipidemia, diabetes, and active smoking—predict CVD risk with about 70% accuracy, additional research on nontraditional CVD risk factors for women is much needed in order to improve women’s cardiovascular health outcomes [7]. The American Heart Association (AHA) recently highlighted the importance of psychological factors on CVD risk emphasizing the need for additional research on modifiable psychological risk factors [8]. Subjective wellbeing and self-rated health are two psychological factors which have been linked to adverse cardiovascular health outcomes [9,10,11,12], particularly among women.

Wellbeing is a multifaceted concept, classified into two different perspectives, hedonic (i.e., subjective) and eudaimonic (i.e., psychological) wellbeing [13]. Based on philosophic origins, eudaimonia, from the Aristotelian perspective, refers to the pursuit of excellence and a good life [14]. Eudaimonic wellbeing or psychological wellbeing deals with optimal human functioning and is conceptualized to consist of six dimensions of positive functioning: self-acceptance, positive relations with others, environmental mastery, autonomy, purpose in life, and personal growth [15]. On the other hand, hedonia, from the Aristippus perspective, refers to the pursuit of pleasure or happiness [14]. Hedonic wellbeing or subjective wellbeing is concerned with happiness and is conceptualized to have three components: (1) life satisfaction, (2) positive affect, and (3) negative affect [14]. Subjective and psychological wellbeing are related, but distinct parts of optimal wellbeing [16]. While the three domains of subjective wellbeing (life satisfaction, positive affect, and negative affect) are all highly correlated with each other, they are separate and distinct components and are recommended to be investigated separately [17]. Each component of subjective wellbeing is potentially influenced by different factors and can also have diverse associations with health outcomes; therefore, research should investigate the separate impact of life satisfaction and affect.

Additionally, sex and age have an interactive effect on subjective wellbeing. Prior to middle adulthood, women are more likely to have higher levels of subjective wellbeing and report being happier; however, after middle age, this reverses with men reporting greater wellbeing than women in older adulthood [18]. Thus, there is a need to examine how subjective wellbeing influences health separately by focusing on women only.

A recent cross-sectional study found higher levels of life satisfaction, one dimension of subjective wellbeing, associated with lower odds of coronary artery calcification [19]. Furthermore, a longitudinal study using data from the Midlife in the United States (MIDUS) study found that individuals with greater life satisfaction and positive affect had a reduced risk for incident cardiometabolic conditions 8–11 years later [20]. Furthermore, the first study [18] focused on a single cardiovascular condition as the outcome, while the latter [19] included diabetes as a cardiometabolic condition. One of the few existing longitudinal studies in women found a composite measure of subjective wellbeing associated with less progression of coronary artery calcification [21]. However, the independent effect of a subjective wellbeing on long-term cardiovascular health among women population is still not well understood.

Self-rated health (SRH) is a valid subjective indicator of health status. The AHA recognizes poor SRH as a risk factor for CVD [22] and as a useful clinical tool for disease risk screening [23]. SRH is a strong predictor of both morbidity and mortality [24,25], and it is significantly associated with CVD [12,26]. Among women with a past history of myocardial ischemia (MI), better self-rated health was associated with a delay of a major CVD event (i.e., MI, stroke, heart failure, or CVD-related death) during their lifetime [27]. In a prospective study of British middle-aged adults, individuals with worse SRH had increased odds for incident cardiovascular events, both fatal and nonfatal, 11 years later. Importantly, SRH remained a significant predictor for cardiovascular events even after controlling for demographic, behavioral, and clinical risk factors, suggesting that it is an independent predictor for cardiovascular health [26]. Although the reliability of self-report is sometimes questioned in the literature, women’s self-rated health has consistently been significantly associated with lifetime cardiovascular disease risk [28]. However, most studies are cross-sectional in design, and the use of longitudinal data is warranted to examine SRH’s predictability of lifetime CVD risk.

In response, the current study capitalized on large-scale, publicly available, longitudinal data from the Midlife in the United States (MIDUS) study and examined independent associations of subjective wellbeing (e.g., life satisfaction, positive affect, and negative affect) and SRH on predicting women’s lifetime CVD risk. The findings of this study may shed light on the utility of self-rated psychological health assessments in evaluating women’s future cardiovascular risk and inform the development of targeted interventions for the women population.

## 2. Materials and Methods

### 2.1. Participants and Procedure

The MIDUS survey is a US national prospective study, spanning nearly 20 years, conducted to understand bio-psychosocial factors influencing health and wellbeing in middle-aged and older adults. Noninstitutionalized English-speaking adults were recruited for the first wave (baseline) using random digit dialing from the 48 contiguous states. Currently, three waves of MIDUS data have been released, with each wave approximately 9 years apart. Wave 1 of the MIDUS study was conducted in 1995–1996, with wave 2 from 2004–2006, and wave 3 from 2013–2014. For all three waves, a telephone interview was conducted and followed by a self-administered questionnaire via mail. The total number of MIDUS participants at wave 1 was 7106; of these, 51.59% (n = 3666) were women. At wave 2, 72.20% of these women were retained (n = 2647). Figure 1 depicts the study sample selection. A total of 1808 women participated in all three waves of the MIDUS study, and the current study excluded 92 participants with missing data on variables of interest, yielding a final sample size of 1716. Collection of MIDUS data was approved by the Institutional Review Board (IRB) at the University of Wisconsin-Madison, and all participants provided informed consent. Since the current study is a secondary analysis of publicly available deidentified data, via the Inter-University Consortium for Political and Social Research website, formal IRB (IRB) approval for this study was waived.

Table 1 presents descriptive statistics of demographic characteristics. The participants’ ages at baseline ranged from 20–74 years (mean 45.73, standard deviation 11.51), and they were predominantly non-Hispanic White (93.6%) and married (70.5%). About one-third (33.9%) held a 4 year college degree or higher, 65% were employed, 17% were current smokers, and 2% had diabetes. The majority of participants rated their current physical health at baseline (wave 1) as either very good (n = 672; 39.2%) or good (n = 568; 33.1%), and their health compared to others as somewhat better (n = 545; 31.8%) or about the same (n = 679; 39.6%).

### 2.2. Study Variables

#### 2.2.1. Covariates 

Covariates included demographics and cardiovascular risk factors at baseline. Demographic factors included age (years), race (non-white vs. white), marital status (not married vs. married), education level (less than high school, high-school degree, some college, 4 year college degree or higher), employment status (not currently working vs. working), household income (US dollars), and pre-existing CVD at wave 1. Household incomes were capped to a maximum value of $300,000. Cardiovascular risk factors included smoking status (former or never smokers vs. current smokers) and diabetes (no diabetes vs. self-reported diabetes or high blood sugar). 

#### 2.2.2. Subjective Wellbeing 

Three separate and distinct components of subjective wellbeing were assessed at each wave: life satisfaction, positive affect, and negative affect. Following recommendations from prior research, life satisfaction and affects were investigated separately [16]. Life satisfaction was measured using a five-item self-report questionnaire [29]. Participants were asked to rate their satisfaction with five dimensions of their lives (overall life, work, health, relationship with spouse/partner, and relationship with children) on an 11-point Likert scale from 0 (worst possible) to 10 (best possible). Scores were calculated as the mean of the five items, where higher scores indicate greater life satisfaction, with adequate internal consistency (Cronbach α, 0.68) using the study data. Participants without spouses and/or children were instructed to leave the items on spouses and children unanswered (blank). When there were missing data on these items, imputed scores were calculated. For those with at least one response on items of spouses and children, scores of answered items on spouses and children were averaged, and the average score was used for nonresponse items. If no items on spouses and children were answered, then a mean score of the remaining spouse/children-unrelated items was used as a score [29].

Positive affect and negative affect were assessed using the six-item positive affect scale (PAS) and six-item negative affect scale (NAS) [30]. Participants were asked the following question, with six positive and six negative emotions as responses: “During the past 30 days, how much of the time did you feel?” Sample emotions included cheerful and extremely happy for the PAS, and nervous and hopeless for the NAS. Items were answered on a five-point Likert scale from 1 (all of the time) to 5 (none of the time) and reverse-coded. PAS and NAS scores were calculated by taking the mean of all six items, where higher scores indicate greater positive and negative emotions. Both the PAS (Cronbach α, 0.91) and the NAS (α 0.86) showed good internal consistency. 

#### 2.2.3. Self-Rated Health (SRH) 

Two types of SRH—physical SRH and health compared to others—were assessed at each wave. Self-rated physical health was measured using a single item on a five-point Likert scale: 1 (poor), 2 (fair), 3 (good), 4 (very good), and 5 (excellent). Greater scores indicate better self-rated physical health. Self-rated health compared to others at your age was assessed at each wave using a single item on a five-point Likert scale: 1 (much better), 2 (somewhat better), 3 (about the same), 4 (somewhat worse), and 5 (much worse). Higher scores indicate worse health compared to others. 

#### 2.2.4. Cardiovascular Conditions 

Cardiovascular disease (CVD) events/conditions were assessed at each wave similar to prior research [31]. Participants self-reported if they had each of the following cardiovascular health problems: stroke, heart trouble suspected/confirmed by doctor, heart attack, angina, high blood pressure (hypertension), valve disease, hole in heart, blocked artery, irregular heartbeat, heart murmur, heart failure, and other heart problem (including hyperlipidemia). Participants at each wave were asked the following question for each of the cardiovascular health problems described above: “In the past 12 months, have you experienced or been treated for … ?”. Participants were classified as having a cardiovascular event/condition if they reported experiencing at least one.

### 2.3. Statistical Methods

Descriptive analyses for continuous (means, standard deviations, and correlations) and categorical (frequencies and percentages) study variables are presented. Chi-square and t-tests were used to examine the association of categorical and continuous predictors with CVD at follow-up waves 2 (9–11 years later) and 3 (17–19 years later). Multiple logistic regressions, models 1–3, for each domain (subjective wellbeing and SRH) were run on CVD conditions 10–19 years later, with covariates. Odds ratios (OR) and 95% confidence intervals (95% CI) are presented as regression results.

For subjective wellbeing models, model 1 was adjusted for baseline covariates including demographics and CVD risk factors (age, race, marital status, education, employment status, income, smoking behavior, diabetes, and pre-existing CVD); model 2 was additionally adjusted for life satisfaction at baseline, and model 3 was fully adjusted with positive and negative affect. For the SRH models, model 1 was the same as subjective wellbeing model 1, whereas model 2 included self-rated physical health at baseline, and model 3 additionally added SRH compared to others. The goodness of fit of logistic models was assessed using the Hosmer–Lemeshow goodness-of-fit test, C-statistics, and AIC (Alkaline information criteria). Smaller values for AIC and greater values for C-statistics indicated a better fit, with C-statistic values over 0.7 considered a good fit. Furthermore, sensitivity analyses were conducted to test goodness of fit of excluding the pre-existing CVD condition from the models.

Lastly, receiver operating characteristic (ROC) curve analyses were conducted, and the highest Youden’s Index (sensitivity + specificity − 1) value was used to determine the optimal cutoff points of subjective wellbeing and SRH measures that discriminate women at risk of CVD at wave 3 from no risk. Two-tailed *p*-values <0.05 were considered statistically significant for all analyses. 

All statistical analyses were conducted in SPSS version 29.0 (SPSS Inc., Chicago, IL, USA).

## 3. Results

Participants reporting cardiovascular conditions significantly increased over time; 22.5% (n = 386) of participants had a cardiovascular condition or health problem at wave 1, which increased to 41.8% (n = 718) at wave 2 (9–11 years later) and 54.4% (n = 933) at wave 3 (17–19 years later). Participants with a CVD condition at waves 2 and 3 were older in age, had less income and a lower education level, were not working at wave 1 and diabetic, and had pre-existing CVD at wave 1 compared those with no CVD (see Table 1).

For both subjective wellbeing and SRH model 1, women who were older, non-white, and had pre-existing CVD had greater odds of reporting a cardiovascular condition about 10 years later at wave 2. On the other hand, only age and pre-existing CVD were significantly associated with CVD conditions 19 years later at wave 3 (Table 2).

For the subjective wellbeing models, participants with higher baseline life satisfaction had a significantly lower risk of a cardiovascular event at 10 years (wave 2) and 19 years (wave 3) later for model 2 (10 years OR 0.86, 95% CI 0.78–0.95; 19 years OR 0.82, 95% CI 0.75–0.90) and model 3 (10 years OR 0.83, 95% CI 0.74–0.94; 19 years OR 0.83, 95% CI 0.74–0.93; Table 2). However, positive and negative affect (subjective wellbeing model 3) were not significantly associated with cardiovascular conditions over time after controlling for all covariates. Model 2 demonstrated an adequate fit based on the Hosmer–Lemeshow test (*p* > 0.05) and was the best model among all three models according to C-statistics and AIC.

For SRH models, individuals with better self-rated physical health had significantly lower odds of having cardiovascular health problems at 10 and 19 years later for model 2 (10 years OR 0.74, 95% CI 0.64–0.85; 19 years OR 0.71, 95% CI 0.62–0.81) and model 3 (10 years OR 0.79, 95% CI 0.68–0.92; 19 years OR 0.74, 95% CI 0.64–0.86; Table 2). Health compared to others was not significantly associated with cardiovascular conditions 10 (wave 2) or 19 (wave 3) years later. SRH model 2 demonstrated the best model fit based on C-statistic and AIC among all three models (Table 2) and was adequate according to the Hosmer–Lemeshow test (*p* > 0.05).

When considering subjective wellbeing and SRH models using wave 2 as baseline in predicting cardiovascular conditions at 9–11 years later at wave 3, the results were similar to the logistic regression using wave 1 as baseline except for health compared to others (Appendix A). Worse health compared to others at wave 2 predicted a higher risk of CVD at wave 3 (OR 1.30, 95% CI: 1.11–1.53; Appendix A).

Sensitivity analyses of excluding existing CVD condition from the models showed similar results of life satisfaction and SRH (Appendix A). Without pre-existing CVD in the models, presence of diabetic condition was significantly associated with increased CVD risk at 10 years and 19 years later (Appendix A).

On the basis of the ROC curve results, the optimum cutoff points—based on the highest Youden index value—for life satisfaction score to discriminate women at a high risk of CVD from a low risk were 7.65 and 8.29 in predicting lifetime risk of CVD at 10 and 19 years, respectively (10 years, sensitivity 64% and specificity 45%; 19 years, sensitivity 43% and specificity 66%; Table 3). For SRH, the optimal cutoff value was 3.50 for physical SRH at both 10 and 19 years (10 years, sensitivity 66% and specificity 55%; 19 years, sensitivity 67% and specificity 52%; Table 3).

## 4. Discussion

The current study found that older age, being non-white, life satisfaction, and self-rated health (SRH) at baseline are independent predictors of women’s cardiovascular health at 10 years and even almost 20 years later. Certain psychological factors, such as life satisfaction and self-rated physical health can have clinical utility as alternatives to major depression in female patients. Life satisfaction is negatively correlated with depression [32], and SRH is a significant predictor of major depression up to 5 years later in females [33]. Thus, both life satisfaction and self-rated health can be used as proxy measures for CVD risk.

The current study findings add to the growing literature examining the association between psychological risk factors and CVD, particularly with a focus on women [18,26]. Prior research using MIDUS data found a greater life satisfaction associated with lower odds of cardiometabolic conditions at 8–11 years later [20]. The current study builds upon past research by finding life satisfaction and self-reported physical health as predictors of a long-term cardiovascular health specifically in women. Indeed, the past research has found sex-differences in the influence of psychological risk factors on cardiovascular health for men and women. Specifically, women with low life satisfaction, but not men, had worse cardiovascular health measures, such as body mass index (BMI), blood pressure, cholesterol, and glucose levels [34].

Previous studies using a composite global measure of wellbeing found that subjective wellbeing was associated with less coronary artery calcification in women [21] and reduced risk of hypertension and dyslipidemia, as well as lower Framingham risk score, in men and women [35]. When looking at specific domains of subjective wellbeing, the current study found only life satisfaction, but not positive or negative affect, to be significantly associated with a long-term cardiovascular health for women. Furthermore, the beneficial influence of life satisfaction was retained after controlling for other CVD risk confounders in the model, suggesting that the domains of subjective wellbeing indeed seem to be separate and distinct constructs. Previous research on positive affect has yielded inconsistent results regarding its association with cardiovascular health over time. While some studies found similar nonsignificant results with cardiovascular health over time [36], others found positive affect to be protective against incident heart disease [37]. In the current study, women reporting a cardiovascular condition/health problem at 10 years and 17–19 years later constantly had significantly lower levels of life satisfaction at baseline. This indicates that life satisfaction may be an independent factor that determines long-term cardiovascular health for women.

Consistent with prior research, worse physical SRH was associated with greater odds of having cardiovascular conditions/events at 10 and almost 20 years later. Health behaviors and clinical risk factors are thought to be potential mechanisms to explain the association between SRH and CVD. Physical SRH remained a significant predictor of cardiovascular health even after controlling for demographics, smoking status, and chronic conditions of diabetes and CVD, thus suggesting its robust association with cardiovascular health in women. These findings suggest that physical SRH may be an independent factor, particularly in women, predicting elevated CVD risk. Indeed, prior research in women with CVD found SRH to be similar in risk magnitude to traditional risk factors (i.e., diabetes and dyslipidemia) [27]. Furthermore, SRH—in conjunction with blood pressure, age, and smoking status—adequately determined CVD risk for women, at least compared to the Framingham model, suggesting that SRH can be a useful tool for clinicians in assessing CVD risk for women [23]. A meta-analysis of prospective studies in Europe and the United States found poor or fair SRH to be associated with cardiovascular-related mortality, suggesting that SRH is valuable for patients with or at risk for CVD [38]. These findings support the utility of physical SRH as an important predictor of cardiovascular health.

Our findings provide robust support for the significant roles of subjective wellbeing and SRH in cardiovascular health for women nearly 20 years later. These findings add to the existing literature and highlight the significance of life satisfaction and self-rated physical health as important psychological factors influencing women’s cardiovascular health. Additionally, the use of a large national dataset, the MIDUS study, which includes adults in early and middle adulthood at baseline, allowed for valuable insight into the impact psychological factors on cardiovascular health during a critical period of life, i.e., middle age, when clinical risk factors start to emerge.

Potential mechanisms linking wellbeing [9] with CVD conditions include both behavioral and biological pathways. Greater wellbeing is thought to lead to more healthy behaviors and fewer unhealth behaviors, thereby impacting one’s cardiovascular health. Wellbeing can strengthen restorative processes, such as physical activity and fruit/vegetable consumption, as well as reduce deteriorative processes and alcohol consumption [10]. Additionally, the protective influence of wellbeing on CVD health is also thought to occur via indirect biological pathways (i.e., inflammation and cortisol) [9].

Using the MIDUS study, our study reports empirical optimal points of life satisfaction and SRH for CVD risk stratification. Women with life satisfaction scores of 8 and below—on scale from 0 to 10—are at a higher risk for lifetime CVD conditions than those who score above 8. Additionally, women reporting their self-rated physical SRH as good, fair, or poor—below a cutoff point of 3.5—have a greater lifetime risk for cardiovascular conditions. Our study’s optimal cutoff points for CVD risk stratification are slightly higher than the previous literature [38]. This may suggest that our study’s population, majority (94%) white women, may over inflate their self-rated wellbeing and physical health, partly attributed to the fact that women tend to be more optimistic and overestimate their health compared to men [39,40]. Additionally, older adults, whites, and those with higher socioeconomic status (SES) have also been found to overestimate their health [41]. Therefore, it requires caution and consideration of the tendency of women, particularly white with higher SES, to overestimate their health, when applying life satisfaction and SRH measures to clinical setting in assessing women’s CVD risk.

The unexpected finding of health compared to others being an independent CVD risk factor when using wave 2 data, but not when using wave 1 data (Appendix A), may be accounted for by an age group difference in participants’ response on health compared to others. Aging may add more precision to the “health compared to others” measure in predicting future cardiovascular health.

Pre-existing CVD condition is a well-established risk factor for future CVD incidences and events among women, which is consistent with the current study models. Our sensitivity analyses of excluding pre-existing CVD condition from the prediction models on future CVD incidences showed diabetes as a significant CVD risk factor (Appendix A), in contrast to our main finding of models with pre-existing CVD, presence of diabetes was not significantly associated with future CVD incidences (Table 3). This may be accounted for by the fact that diabetes is one of the known CVD risk factors [42,43] and is significantly correlated with pre-existing CVD conditions (wave 1 χ^2^ = 40.56, *p* < 0.001; wave 2 χ^2^ = 80.74, *p* < 0.001; wave 3 χ^2^ =72.37, *p* < 0.001). Including both pre-existing CVD and diabetes conditions may result in multicollinearity; the study used both AIC and C statistics and tested the model fit of inclusion of these covariates. Following model fit tests, the study’s current models with pre-existing CVD condition as a baseline covariate showed a better fit of models than without it.

This study had several limitations. First, study data were self-reported, particularly cardiovascular disease diagnoses without adjudication; thus, they may be biased, although the self-report of medical conditions has been found to be reliable [44]. Future research should incorporate clinical assessments of cardiovascular conditions. Second, the generalizability of the study results is limited due to the overwhelming proportion of white women and underrepresentation of racial and ethnic minority groups. For example, reported cutoff points of life satisfaction and SRH are empirically driven and may be specific to our study data. Future studies with more diverse, representative data are warranted for optimal cutoff points of subjective wellbeing measures. Third, cardiovascular health conditions/problems at all three waves were serially correlated with each other. Future studies are warranted to examine serial correlations of subjective wellbeing and self-rated physical health with CVD conditions to determine its causation direction and residual effects.

### Implications

Despite these limitations, the current study contributes to the existing literature by finding the significant impact of life satisfaction and SRH on long-term cardiovascular health up to 20 years later in women. Both life satisfaction and SRH are modifiable CVD risk factors and can serve as an important point for public health interventions in reducing the burden of CVD. A meta-analysis [45] showed that interventions—mindfulness, cognitive approaches, and other psychological interventions—were particularly effective in increasing subjective (hedonic) wellbeing. Additionally, a gratitude intervention in women found that participants had greater life satisfaction and lower diastolic blood pressure [46], while another eHealth intervention in patients with CVD was found to improve subjective wellbeing [47]. Furthermore, healthcare providers may benefit from considering patients’ subjective wellbeing and SRH when evaluating and managing those at risk for cardiovascular disease. Adopting a more comprehensive approach considering both physical and psychological aspects of cardiovascular health may improve the identification of women who are at risk and reduce the overall burden of CVD in women. In particular, a single-item SRH measure is a convenient and low-cost assessment that could be used as a quick screening tool for cardiovascular health.

## 5. Conclusions

The present study supports that aging, non-white racial background, and psychological risk factors (subjective wellbeing and SRH) are associated with cardiovascular health in women over time. Greater levels of subjective wellbeing, life satisfaction, and better self-rated physical health were associated with lower odds of cardiovascular events up to almost 20 years later for adult women. Understanding the role of psychological factors in cardiovascular health is crucial for developing effective public health interventions. Future research is needed to validate the current study’s findings in a more diverse population including racial and ethnic minority women.

## Figures and Tables

**Figure 1 ijerph-20-06380-f001:**
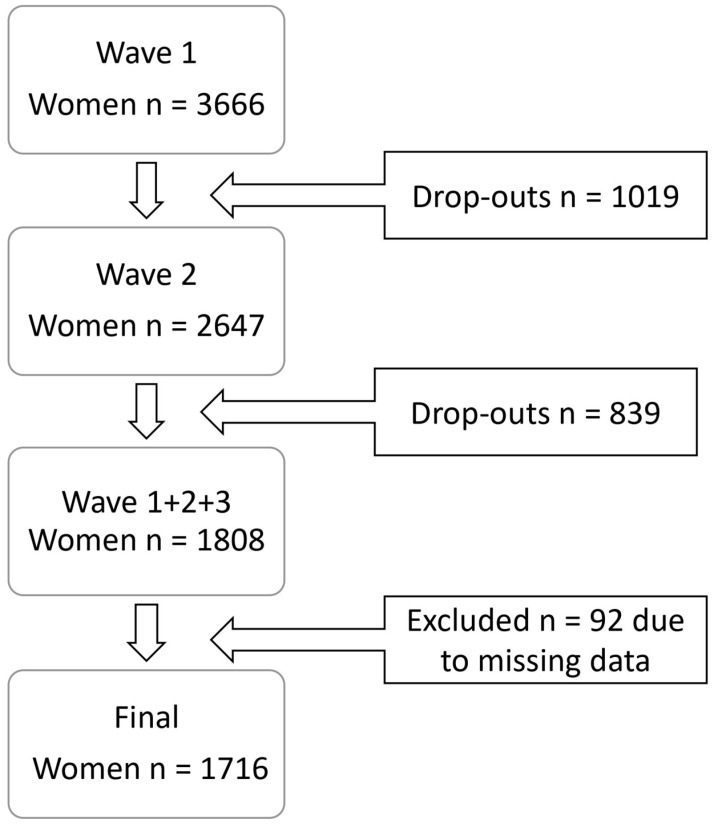
Study sample selection.

**Table 1 ijerph-20-06380-t001:** Baseline characteristics of study participants (n = 1716).

Characteristic at Wave 1		CVD Conditions (Wave 2)			CVD Conditions (Wave 3)		
	Mean (SD) or n (%)	No (n = 998)	Yes (n = 718)	*p*-Value	No (n = 783)	Yes (n = 933)	*p*-Value
Age (years)	45.73 (11.51)	42.69 (10.85)	49.96 (11.07)	<0.001	41.76 (10.55)	49.07 (11.23)	<0.001
Income (US dollars)	76,967 (66,040)	81,668 (67,632)	70,438 (65,123)	<0.001	83,526 (68,752)	71,463 (63,188)	<0.001
Race				0.002			0.053
White	1607 (93.6%)	950 (95.2%)	657 (91.5%)		743 (94.9%)	864 (92.6%)	
Non-white	109 (6.4%)	48 (4.8%)	61 (8.5%)		40 (5.1%)	69 (7.4%)	
Education				<0.001			<0.001
Less than high school	105 (6.1%)	42 (4.2%)	63 (8.8%)		32 (4.1%)	73 (7.8%)	
High-school diploma/GED	486 (28.3%)	269 (27.0%)	217 (30.2%)		207 (26.4%)	279 (29.9%)	
Some college	543 (31.6%)	303 (30.4%)	240 (33.4%)		230 (29.4%)	313 (33.5%)	
College degree or higher	582 (33.9%)	384 (38.5%)	198 (27.6%)		314 (40.1%)	268 (28.7%)	
Marital status				0.435			0.345
Married	1210 (70.5%)	711 (71.2%)	499 (69.5%)		561 (71.6%)	649 (69.6%)	
Not married	506 (29.5%)	287 (28.8%)	219 (30.5%)		222 (28.4%)	284 (30.4%)	
Employment status				0.005			0.006
Currently working	1106 (64.5%)	671 (67.2%)	435 (60.6%)		532 (67.9%)	574 (61.5%)	
Not currently working	610 (35.5%)	327 (32.8%)	283 (39.4%)		251 (32.1%)	359 (38.5%)	
Smoking status				0.459			0.275
Current	297 (17.3%)	167 (16.7%)	130 (18.1%)		127 (16.2%)	170 (18.2%)	
Former/never	1419 (82.7%)	831 (83.3%)	588 (81.9%)		656 (83.8%)	763 (81.8%)	
Diabetes/high blood sugar				<0.001			<0.001
Yes	34 (2.0%)	10 (1.0%)	24 (3.3%)		6 (0.8%)	28 (3.0%)	
No	1682 (98%)	988 (99.0%)	694 (96.7%)		777 (99.2%)	905 (97.0%)	
Pre-existing CVD				<0.001			<0.001
Yes	386 (22.5%)	51 (5.1%)	335 (46.7%)		42 (5.4%)	344 (36.9%)	
No	1330 (77.5%)	947 (94.9%)	383 (53.3%)		741 (94.6%)	589 (64.1%)	
Life satisfaction	7.81 (1.22)	7.88 (1.18)	7.71 (1.27)	0.004	7.93 (1.18)	7.72 (1.25)	<0.001
Positive affect	3.39 (0.73)	3.42 (0.70)	3.35 (0.76)	0.063	3.45 (0.70)	3.34 (0.74)	0.002
Negative affect	1.55 (0.63)	1.52 (0.60)	1.59 (0.67)	0.032	1.50 (0.58)	1.59 (0.66)	0.005
Physical self-rated health	3.64 (0.92)	3.82 (0.88)	3.39 (0.91)	<0.001	3.87 (0.88)	3.45 (0.90)	<0.001
Health compared to others	2.28 (0.89)	2.22 (0.86)	2.37 (0.93)	<0.001	2.21 (0.86)	2.34 (0.91)	0.002

Abbreviations. SD = standard deviation; CVD = cardiovascular disease. *p*-Values are from chi-square test for categorical variables and *t*-test for continuous variables.

**Table 2 ijerph-20-06380-t002:** Logistic regression analysis, odds ratios (95% confidence intervals), for subjective wellbeing and self-rated health (wave 1) at baseline predicting cardiovascular conditions 10 years (wave 2) and 19 years later (wave 3).

Variables at Wave 1	CVD Conditions (Wave 2) OR (95% CI)			CVD Conditions (Wave 3) OR (95% CI)		
	Model 1	Model 2	Model 3	Model 1	Model 2	Model 3
**Subjective Wellbeing Model**						
Age	1.05 (1.04, 1.06)	1.06 (1.04, 1.07)	1.05 (1.04, 1.06)	1.05 (1.04, 1.06)	1.06 (1.05, 1.07)	1.06 (1.05, 1.07)
White vs. non-white	0.45 (0.29, 0.71)	0.46 (0.29, 0.73)	0.46 (0.29, 0.72)	0.62 (0.39, 0.97)	0.64 (0.41, 1.00)	0.63 (0.40, 1.00)
Married vs. not married	0.92 (0.70, 1.21)	0.97 (0.74, 1.28)	0.97 (0.73, 1.27)	0.93 (0.72, 1.19)	0.99 (0.77, 1.29)	0.99 (0.77, 1.29)
<HS degree	---	---	---	---	---	---
HS degree	0.87 (0.52, 1.48)	0.87 (0.51, 1.48)	0.86 (0.51, 1.47)	0.92 (0.55, 1.54)	0.91 (0.54, 1.54)	0.92 (0.54, 1.54)
Some college	0.95 (0.56, 1.60)	0.93 (0.55, 1.58)	0.92 (0.54, 1.56)	1.05 (0.62, 1.76)	1.02 (0.61, 1.72)	1.03 (0.61, 1.73)
≥4 year college	0.66 (0.39, 1.14)	0.65 (0.38, 1.13)	0.65 (0.37, 1.11)	0.74 (0.44, 1.25)	0.72 (0.43, 1.23)	0.73 (0.43, 1.23)
Not working vs. working	1.18 (0.92, 1.52)	1.18 (0.91, 1.51)	1.16 (0.90, 1.49)	1.12 (0.89, 1.42)	1.11 (0.88, 1.40)	1.11 (0.88, 1.41)
Income (US dollars)	1.00 (1.00, 1.00)	1.00 (1.00, 1.00)	1.00 (1.00, 1.00)	1.00 (1.00, 1.00)	1.00 (1.00, 1.00)	1.00 (1.00, 1.00)
Current vs. former/never smoker	1.10 (0.80, 1.51)	1.07 (0.78, 1.47)	1.08 (0.79, 1.48)	1.18 (0.88, 1.58)	1.13 (0.84, 1.53)	1.13 (0.84, 1.52)
Diabetic vs. nondiabetic	1.11 (0.42, 2.96)	1.04 (0.39, 2.75)	1.04 (0.39, 2.78)	1.79 (0.63, 5.08)	1.62 (0.57, 4.63)	1.61 (0.57, 4.61)
Pre-existing CVD vs. no CVD	13.86 (9.96, 19.28)	13.30 (9.57, 18.56)	13.51 (9.69, 18.83)	8.12 (5.73, 11.51)	7.72 (5.44, 10.95)	7.69 (5.42, 10.92)
Life satisfaction		0.86 (0.78, 0.95)	0.83 (0.74, 0.94)		0.82 (0.75, 0.90)	0.83 (0.74, 0.93)
Positive affect			0.96 (0.75, 1.22)			0.94 (0.77, 1.16)
Negative affect			1.08 (0.86, 1.34)			0.98 (0.77, 1.24)
Hosmer–Lemeshow (*p*-value)	0.10	0.16	0.14	0.85	0.83	0.94
C-statistic	0.80	0.80	0.80	0.80	0.80	0.80
AIC	1808.71	1801.51	1804.48	1985.10	1969.92	1973.60
**Self-Rated Health Model**						
Age	1.05 (1.04, 1.06)	1.05 (1.04, 1.06)	1.06 (1.04, 1.07)	1.05 (1.04, 1.06)	1.05 (1.05, 1.07)	1.06 (1.05, 1.07)
White vs. non-white	0.45 (0.29, 0.71)	0.51 (0.32, 0.80)	0.49 (0.31, 0.78)	0.62 (0.39, 0.97)	0.71 (0.46, 1.11)	0.69 (0.44, 1.10)
Married vs. not married	0.92 (0.70, 1.21)	0.95 (0.72, 1.24)	0.93 (0.71, 1.22)	0.93 (0.72, 1.19)	0.96 (0.77, 1.25)	0.95 (0.73, 1.22)
<HS degree	---	---	---	---	---	---
HS degree	0.87 (0.52, 1.48)	0.96 (0.57, 1.64)	0.94 (0.55, 1.60)	0.92 (0.55, 1.54)	1.03 (0.52, 1.38)	1.01 (0.60, 1.71)
Some college	0.95 (0.56, 1.06)	1.08 (0.64, 1.84)	1.06 (0.62, 1.81)	1.05 (0.62, 1.76)	1.23 (0.65, 1.72)	1.21 (0.71, 2.05)
≥4 year college	0.66 (0.39, 1.14)	0.78 (0.45, 1.36)	0.77 (0.44, 1.33)	0.74 (0.44, 1.25)	0.90 (0.48, 1.30)	0.89 (0.52, 1.52)
Not working vs. working	1.18 (0.92, 1.52)	1.23 (0.95, 1.58)	1.23 (0.96, 1.59)	1.12 (0.89, 1.42)	1.16 (0.87, 1.37)	1.16 (0.91, 1.47)
Income (US dollars)	1.00 (1.00, 1.00)	1.00 (1.00, 1.00)	1.00 (1.00, 1.00)	1.00 (1.00, 1.00)	1.00 (1.00, 1.00)	1.00 (1.00, 1.00)
Current vs. Former/never smoker	1.10 (0.80, 1.51)	1.05 (0.76, 1.44)	1.04 (0.76, 1.43)	1.18 (0.88, 1.58)	1.10 (0.85, 1.49)	1.10 (0.82, 1.48)
Diabetic vs. nondiabetic	1.11 (0.42, 2.96)	0.93 (0.34, 2.52)	0.91 (0.34, 2.48)	1.79 (0.63, 5.08)	1.51 (0.92, 6.34)	1.49 (0.51, 4.41)
Pre-existing CVD vs. no CVD	13.86 (9.96, 19.28)	12.33 (8.83, 17.22)	12.18 (8.72, 17.01)	8.12 (5.73, 11.51)	7.04 (4.95, 10.03)	6.96 (4.89, 9.91)
Physical SRH		0.74 (0.64, 0.85)	0.79 (0.68, 0.92)		0.71 (0.62, 0.80)	0.74 (0.64, 0.86)
Health compared to others			1.14 (0.98, 1.34)			1.10 (0.95, 1.28)
Hosmer–Lemeshow (*p*-value)	0.10	0.72	0.16	0.85	0.83	0.80
C-statistic	0.81	0.81	0.80	0.80	0.80	0.80
AIC	1804.55	1787.96	1787.19	1980.94	1954.37	1954.65

Abbreviations. HS = high school; CVD = cardiovascular disease; SRH = self-rated health; AIC = Akaike information criteria. Values are statistically significant at the α = 0.05 significance level when 95% CI does not contain 1. A Hosmer–Lemeshow *p*-value > 0.05 indicates a good model fit.

**Table 3 ijerph-20-06380-t003:** Optimal cutoff points of life satisfaction and physical self-rated heath (SRH) scores to predict cardiovascular disease (CVD) at 10 years (wave 2) and 17–19 years later (wave 3).

Subjective Wellbeing Measures	Cutoff Values	AUC (95% CI)	Sensitivity	Specificity	Youden Index	*p*-Value
**CVD at 10 years (Wave 2)**						
Life satisfaction	7.65	0.541 (0.513, 0.569)	0.636	0.450	0.086	0.004
Physical SRH	3.50	0.630 (0.603, 0.657)	0.660	0.554	0.215	0.000
**CVD at 17–19 years (Wave 3)**						
Life satisfaction	8.29	0.550 (0.522, 0.577)	0.434	0.659	0.093	0.000
Physical SRH	3.50	0.626 (0.600, 0.652)	0.674	0.517	0.191	0.000

Abbreviations. AUC = area under the receiver operator characteristic (ROC) curve; CI = confidence interval; CVD = cardiovascular disease; SRH = self-rated health. The *p*-value is from the ROC curve analysis. Receiver operator characteristic (ROC) curve analysis was conducted to calculate the sensitivity and specificity of subjective wellbeing measures (wave 1) predicting cardiovascular conditions at waves 2 and 3. Model 2 was selected for both life satisfaction and physical SRH models.

## Data Availability

The current study used publicly available data at https://www.icpsr.umich.edu/web/pages/NACDA/midus.html (accessed on 16 December 2022).

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
