# Peer review of "The Effects of Subjective Wellbeing and Self-Rated Health on Lifetime Risk of Cardiovascular Conditions in Women"

_ijerph, 2023, doi:10.3390/ijerph20146380_

Round 1

Reviewer 1 Report

In this work, the authors investigated the association between subjective well-being and self-rated health at baseline and the risk of CVD occurrence in 10-19 years using a longitudinal study data, the Midlife in the United States study data. The authors built multiple logistic regression models and identified higher level of life satisfaction and self-rated health predict lower cardiovascular disease risk.

Suggestions and questions:

1.     Two of the traditional CVD risk factors, hypertension and hyperlipidemia, were not included in this analysis. These risk factors could confound the association between life satisfaction, SRH and CVD. Is this due to data limitation? Have the authors considered including them in the analysis?

2.     The authors discussed that life satisfaction, positive affect, and negative affect are usually highly correlated, which could potentially lead to multicollinearity. Is this the case in your study? Have the authors considered performing a variable selection?

3.     The study only included women who participated in all three waves of the MIDUS study. What percentage of women participated in all three waves? What are the common reasons for dropout? When some participates dropout in either wave 2 or 3, they can still be included in the analysis for wave 3 or 2 models. What’s the rational of excluding these subjects from the analysis?

4.     Table 1, I expect that income is a right-skewed variable. In that case, I suggest reporting median and IQR, instead of mean and SD.

5.     Table 1, Does the summary statistics at wave 2 and 3 represent the statistics of variables at baseline? I suggest adding a footnote to clarify.

6.     Line 150 to 156, it’s not quite clear to me when participates reported that they have at least one cardiovascular health problem, it is during which period of time? For example, at wave 3 measurement, if a patient reported having cardiovascular issue, does it mean she had a problem from wave 2 to wave 3 measurement?

7.     Line 184, typo? n=386 is inconsistent with Table 1, where n=385.

8.     Line 191-192, the conclusion is not accurate, as from Table 2, race is also significant for model 1.

9.     Table 2, I suggest add the Hosmer-Lemeshow p-value to the table as well.

10.  Line 209, “19 years OR 0.71, 95% CI 0.62-0.809”, 0.809 should also be rounded to two decimal places.

11.  Line 215-217, health compared to others is significant using wave 2 as baseline in predicting wave 3 status. This is different from the models using wave 1 as baseline. Is there any hypothesis why this variable is significant when using wave 2 as baseline, but not when using wave 1 as baseline?

12.  Table 3 does not seem to be correct, Youden’s Index = sensitivity + specificity - 1. In this Table, the index > 0 but sensitivity + specificity < 1. In addition, the AUC, sensitivity and specificity are quite low here, many of them are less than 0.5 (worse than a random guess), why is this the case? Do I misunderstand anything here?

Author Response

Thank you for your comments and feedback. We have addressed reviewers’ concerns and feedback in the revised manuscript. Please find point-by-point responses attached and reviewers’ comments are in Italics

Reviewer #1:
Suggestions and questions:

  1. Two of the traditional CVD risk factors, hypertension and hyperlipidemia, were not included in this analysis. These risk factors could confound the association between life satisfaction, SRH and CVD. Is this due to data limitation? Have the authors considered including them in the analysis?

Response: Both hypertension and hyperlipidemia were included as CVD conditions. We rewrote the description of CVD inclusion criteria for clarification with adding hypertension next to high blood pressure and other heart problems (including hyperlipidemia) in the methods section. Hypertension was asked separately while hyperlipidemia was included in “other heart problem.”

“Participants self-reported if they had each of the following cardiovascular health problems: stroke, heart trouble suspected/confirmed by doctor, heart attack, angina, high blood pressure (hypertension), valve disease, hole in heart, blocked artery, irregular heartbeat, heart murmur, heart failure, and other heart problem (including hyperlipidemia).

  1. The authors discussed that life satisfaction, positive affect, and negative affect are usually highly correlated, which could potentially lead to multicollinearity. Is this the case in your study? Have the authors considered performing a variable selection?

Response: The authors used Akaike Information Criteria (AIC) and C statistics (AUC Area under the ROC curve) for a model selection. We ran three models—models 1 to 3 –to deal with high correlations of measures of life satisfaction, positive and negative affect. We added subjective well-being measures one at a time to the model  and checked model fit of each additional subjective well-being measure as a covariate.  Model 1 served as a basic model including covariates of demographic variables and other known CVD risk factors, Model 2 added life satisfaction to Model 1, and Model 3 is the final model with all covariates, Model 2 +  positive and negative affect.

In response, we added the following Table S2 to show correlations among life satisfaction, positive and negative affect. Pearson correlation 0.5 is considered as weak but it needs caution to add these as covariates to the model simultaneously. Thus, the study used AIC and C statistics  as criteria for the model selection.  The smaller AIC values are better fit, we considered Models 2 and 3 for a sensitivity test.  According to sensitivity tests, adding all three covariates, life satisfaction, positive affect, and negative affect, is acceptable—smaller AIC values (less than 10 difference) and VIF <2.

Table S2. Correlations for Life satisfaction, positive affect, and negative affect

Life Satisfaction

Negative Affect

Positive Affect

VIF

Life Satisfaction

1.00

---

---

1.62

Negative Affect

-.50 (p<.001)

1.00

---

1.83

.56 (p<.001)

-.64 (p<.001)

1.00

1.96

Notes.  VIF = Variance Information Factor

  1. VIF >1 indicates multicollinearity and 10 is considered as high.

Source:

James, G., Witten, D., Hastie, T., & Tibshirani, R. (2013). An introduction to statistical learning (Vol. 112, p. 18). New York: springer.

  1. The study only included women who participated in all three waves of the MIDUS study. What percentage of women participated in all three waves? What are the common reasons for dropout? When some participates dropout in either wave 2 or 3, they can still be included in the analysis for wave 3 or 2 models. What’s the rational of excluding these subjects from the analysis?

In response, we added figure 1 for the details of the sample selection and attritions in three waves and the following sentence for clarification in the methods section.

“The total number of MIDUS participants at wave 1 was n=7,106 and of these, 51.59% (n=3,666) were women. At wave 2, 72.20% of these women were retained (n=2,647).

  1. Table 1, I expect that income is a right-skewed variable. In that case, I suggest reporting median and IQR, instead of mean and SD.

In response, median and IQR of income for all 3 waves (wave 1 32,500-92,500; wave 2 31,250-87,500; wave 3 35,000-88,375) were added to Table S1.  Please see the next comment 5 for adding Table S1.

  1. Table 1, Does the summary statistics at wave 2 and 3 represent the statistics of variables at baseline? I suggest adding a footnote to clarify.

In response, we added supplementary Table S1.  Table S1 depicts summary statistics of characteristics at waves 1, 2, and 3 and test statistics of characteristics comparisons of waves 2 and 3 to wave 1. We added summary of statistics as notes.

  1. Line 150 to 156, it’s not quite clear to me when participates reported that they have at least one cardiovascular health problem, it is during which period of time? For example, at wave 3 measurement, if a patient reported having cardiovascular issue, does it mean she had a problem from wave 2 to wave 3 measurement?

It was a 12-month period and in response, we added a sentence to clarify in the methods section:

Participants at each wave were asked “In the past twelve months, have you experienced or been treated ..?” for each of the cardiovascular health problems described above.

  1. Line 184, typo? n=386 is inconsistent with Table 1, where n=385.

In response, we fixed a typo in the table to n=386 (22.5%).

  1. Line 191-192, the conclusion is not accurate, as from Table 2, race is also significant for model 1.

In response, we added the following underscored terms in the sentences of “The current study found that older age, being non-white, subjective well-being… “ in the beginning of discussion section and “The present study supports aging, non-white racial  background, psychological risk factors…” in the beginning of the conclusions section.

  1. Table 2, I suggest add the Hosmer-Lemeshow p-value to the table as well.

In response, we have added Hosmer-Lemeshow p-values for all models in table 2. All models have  Hosmer-Lemeshow (P value) > 0.05, which  indicates a good model fit.

  1. Line 209, “19 years OR 0.71, 95% CI 0.62-0.809”, 0.809 should also be rounded to two decimal places.

In response, we have rounded the upper 95% CI to 0.81. it is stated as “19 years OR 0.71, 95% CI 0.62-0.81.”

  1. Line 215-217, health compared to others is significant using wave 2 as baseline in predicting wave 3 status. This is different from the models using wave 1 as baseline. Is there any hypothesis why this variable is significant when using wave 2 as baseline, but not when using wave 1 as baseline?

In response, we added a sentence in a results section.

“..except for health compared to others (Supplementary Materials Table S3). Worse health compared to others at wave 2 predicted a higher risk of CVD at wave 3 (OR 1.30, 95% CI: 1.11, 1.53; Table S3).”

Also we added new discussion.

“The unexpected finding, health compared to others was an independent CVD risk factor when using wave 2 data, not when using wave 1 data (Table S3), may be accounted for by an age group difference in participants’ response on health compared to others. Aging may add more precision of “health compared to others” measure in predicting the future cardiovascular health.”

  1. Table 3 does not seem to be correct, Youden’s Index = sensitivity + specificity - 1. In this Table, the index > 0 but sensitivity + specificity < 1. In addition, the AUC, sensitivity and specificity are quite low here, many of them are less than 0.5 (worse than a random guess), why is this the case? Do I misunderstand anything here?

In response, we have revised table 3. 

Reviewer 2 Report

IJERPH-2396680 “The Effects of Subjective Well-Being and Self-Rated Health on Lifetime Risk of Cardiovascular Disease Events in Women”

The authors investigate relationships between subjective well-being and self-rated health on CVD conditions among women in MIDUS. I appreciate the aim to study the relationship between psychological factors and cardiovascular health, however, I had several concerns about the study and analysis. Comments and suggestions are provided below.

Major Comments

1. The authors do not accurately define hedonic vs. eudaimonic well-being. For example, both types of well-being are subjective. Please see extensive work by Carol Ryff and colleagues regarding what each of these terms mean.

2. This is a study using self-reported outcomes. There is a known bias in who receives diagnoses, often with persons from underrepresented racial/ethnic background less likely to receive these diagnoses (including due to lack of access to health care or medical insurance). Thus, even though this study involves longitudinal data, the use of self-reported CVD outcomes is a notable limitation. The authors need to add more details in the limitations about this, as well as providing a citation for their statement that use of self-reported medical conditions is reliable. Moreover, given this limitation, I advise that the authors tone down language in the abstract and across the text that “measures of subjective well-being … can be used as clinical tools for prevention.” Although these measures are certainly useful, it is a leap to say they can be used for prevention (and, what exactly is meant by “prevention”)? Furthermore, in the Discussion, the authors note that “psychological factors are modifiable” – is there data to suggest that life satisfaction and/or SRH could be modified?

3. I have concerns about using pre-existing CVD as a covariate. If the goal is to use well-being measures to predict CVD conditions over time, it seems imperative to exclude individuals with CVD conditions at baseline. Might it not be the case that those who had pre-existing CVD at baseline are actually driving the results, or that those with pre-existing CVD actually report worse well-being over time?

4. The authors use a single-item measure of self-rated health that is specific to the MIDUS study, which should be discussed as a limitation. Relatedly, I was not sure what to make of the “optimal cut-off points” analyses for life satisfaction and SRH. While I can appreciate the clinical value in having cut-off points, they are using an extremely narrow sample of predominantly white, educated women. Thus, the cut-off analyses are not extremely relevant to samples outside of this narrow group.

5. Throughout the text, the authors refer to “subjective well-being” but it is not clear what they mean in different places, as they used life satisfaction, positive affect, and negative affect. Please be clear what components, exactly, you are referring to, in the tables and text rather than using this global term.

6. Given the prior work done on well-being and CVD, including from MIDUS, I am not sure what the current work adds to the literature. It is not sufficient to state that you are using longitudinal data when the prior studies also used longitudinal data (e.g., citation 16). Moreover, I found the rationale to focus solely on women lacking. In the Intro, the authors describe how prior work in this area did not stratify by sex. However, the authors here do not stratify by sex, rather, they just choose to look at women, without a solid justification for why.

7. For the life satisfaction measure, how did the authors deal with people who did not report having a spouse/partner or children?

Minor Comments

8. The term “cardiovascular disease events” in the title is misleading – typically that is used to describe heart attacks and stroke. Would use the term “cardiovascular conditions” like it is described in the text.

9. Citations are needed in line 32 and lines 45-48 of the Intro.

10. In Table 1, please report the breakdown of race/ethnicity rather than simply stating “% non-White”.

11. The tables were difficult to read, particularly column 1, and the text spilling over onto multiple lines in Table 2.

12. The Discussion would benefit from discussion of hypothesized mechanisms linking well-being and CVD conditions.

Author Response

Thank you for your comments and feedback. We have addressed reviewers’ concerns and feedback in the revised manuscript. Please find point-by-point responses attached and reviewers’ comments are in Italics

Review #2:

Comments and Suggestions for Authors

IJERPH-2396680 “The Effects of Subjective Well-Being and Self-Rated Health on Lifetime Risk of Cardiovascular Disease Events in Women”

The authors investigate relationships between subjective well-being and self-rated health on CVD conditions among women in MIDUS. I appreciate the aim to study the relationship between psychological factors and cardiovascular health, however, I had several concerns about the study and analysis. Comments and suggestions are provided below.

Major Comments

  1. The authors do not accurately define hedonic vs. eudaimonic well-being. For example, both types of well-being are subjective. Please see extensive work by Carol Ryff and colleagues regarding what each of these terms mean.

Response:

Thank you for your comment. While both are ‘subjective’ the commonly used terms for hedonic well-being is subjective well-being and for eudaimonic well-being is psychologically well-being. Further clarification has been added in the introduction (lines 45-63).

“Well-being is a multifaceted concept and historically categorized into two unique perspectives, hedonic (i.e. subjective) and eudaimonic (i.e. psychological) well-being [11]. Based on philosophic origins, eudaimonia, from the Aristotelian perspective, refers to the pursuit of excellence and a good life [12]. Eudaimonic well-being or psychological well-being deals with optimal human functioning and is conceptualized to consist of six dimensions of positive functioning –self-acceptance, positive relations with others, environmental mastery, autonomy, purpose in life, and personal growth (14). On the other hand, hedonia, from the Aristippus perspective, refers to the pursuit of pleasure or happiness [12]. Hedonic well-being or subjective well-being which is concerned with happiness and is conceptualized to have three components, 1) life satisfaction, 2) positive affect, and 3) negative affect [12]. Subjective and psychological well-being are related, but distinct parts of optimal well-being [15].”

  1. Ryff CD, Keyes CL. The structure of psychological well-being revisited. J Pers Soc Psychol. 1995;69(4):719-727. doi:10.1037//0022-3514.69.4.719
  2. Keyes CL, Shmotkin D, Ryff CD. Optimizing well-being: the empirical encounter of two traditions. J Pers Soc Psychol. 2002;82(6):1007-1022.
  3. This is a study using self-reported outcomes. There is a known bias in who receives diagnoses, often with persons from underrepresented racial/ethnic background less likely to receive these diagnoses (including due to lack of access to health care or medical insurance). Thus, even though this study involves longitudinal data, the use of self-reported CVD outcomes is a notable limitation. The authors need to add more details in the limitations about this, as well as providing a citation for their statement that use of self-reported medical conditions is reliable.

In response, we added the following sentences in limitations with citations.

“First, study data were self-reported, in particular, cardiovascular events without adjudication, and thus may be biased, although the self-report of medical conditions has been found to be reliable [42].”

Citations:

  1. Martin LM, Leff M, Calonge N, Garrett C, Nelson DE. Validation of self-reported chronic conditions and health services in a managed care population. Am J Prev Med. 2000;18(3):215-218. doi:10.1016/s0749-3797(99)00158-0

Moreover, given this limitation, I advise that the authors tone down language in the abstract and across the text that “measures of subjective well-being … can be used as clinical tools for prevention.” Although these measures are certainly useful, it is a leap to say they can be used for prevention (and, what exactly is meant by “prevention”)?

In response, we changed to “..additional CVD screening tools” in Abstract.

Furthermore, in the Discussion, the authors note that “psychological factors are modifiable” – is there data to suggest that life satisfaction and/or SRH could be modified?

In response, we clarified the sentences adding more explanations and citations in Implications section.

“Both life satisfaction and SRH are modifiable CVD risk factors and can serve as an important point for public health interventions in reducing the burden of CVD. A meta-analysis [43] – were particularly effective in increasing subjective (hedonic) well-being. Additionally, a gratitude intervention in women found participants had greater life satisfaction and lower diastolic blood pressure [44], while another eHealth intervention in patients with CVD was found to improve subjective well-being [45].”

  1. Sakuraya, A.; Imamura, K.; Watanabe, K.; Asai, Y.; Ando, E.; Equchi, H.; Nishida, N.; Kobayashi, Y.; Arima, H.; Iwanage, M.; et al. What Kind of Intervention Is Effective for Improving Subjective Well-Being Among Workers? A Systematic Review and Meta-Analysis of Randomized Controlled Trials. Frontier Psychology 2020, 11, 528656. doi:10.3389/fpsyg.2020.528656
  2. Jackowska, M.; Brown, J.; Ronaldson, A.; Steptoe, A. The Impact of a Brief Gratitude Intervention on Subjective Well-Being, Biology and Sleep. Journal of Health Psychology 2016, 21, 2207-2217. doi:10.1177/1359105315572455
  3. Farhane-Medina, N.Z.; Castillo-Mayén, R.; Luque, B; Rubio, S.J.; Gutierrez-Domingo, T.; Cuadrado, E.; Arenas, A.; Tabernero, C. A Brief mHealth-Based Psychological Intervention in Emotion Regulation to Promote Positive Subjective Well-Being in Cardiovascular Disease Patients: A Non-Randomized Controlled Trial. Healthcare (Basel) 2022, 10, 1640. doi:10.3390/healthcare10091640

  1. I have concerns about using pre-existing CVD as a covariate. If the goal is to use well-being measures to predict CVD conditions over time, it seems imperative to exclude individuals with CVD conditions at baseline. Might it not be the case that those who had pre-existing CVD at baseline are actually driving the results, or that those with pre-existing CVD actually report worse well-being over time?

Response: To test whether life satisfaction and SRH are independent factors associated with the future CVD conditions, we need to control other known risk factors associated with the future CVD conditions.  Existing CVD condition is a well established risk factor for the future CVD incidences in cardiology literature. While not presented in the current manuscript, the study conducted sensitivity analyses without the existing CVD condition model (Table S4) and found the similar results. Please find the following table for results without pre-existing CVD condition, Table S4.

  1. The authors use a single-item measure of self-rated health that is specific to the MIDUS study, which should be discussed as a limitation. Relatedly, I was not sure what to make of the “optimal cut-off points” analyses for life satisfaction and SRH. While I can appreciate the clinical value in having cut-off points, they are using an extremely narrow sample of predominantly white, educated women. Thus, the cut-off analyses are not extremely relevant to samples outside of this narrow group.

Response: We agree with the reviewer’s comment on that cut off points were empirically driven and rather limited to the current study data. However, the cut-off points reported in the current study are comparable to other studies, thus, and the current study cut off points can be used to create optimal cut-off points of subjective well-being measures for the general population along with empirically driven cut off points from different population and data sets.

We pointed out that generalizability of all study results is a major limitation. In response, we added the following sentences as limitation to address reviewer’s concern.

“For example, reported cut-off points of life satisfaction and SRH are empirically driven and may be specific to our study data. The future study with more diverse, representative data is warranted for optimal cutoff points of subjective well-being measures.”

  1. Throughout the text, the authors refer to “subjective well-being” but it is not clear what they mean in different places, as they used life satisfaction, positive affect, and negative affect. Please be clear what components, exactly, you are referring to, in the tables and text rather than using this global term.

 In response, we replaced “subjective well-being” to life satisfaction, positive affect, and negative affect throughout the manuscript as appropriate.

  1. Given the prior work done on well-being and CVD, including from MIDUS, I am not sure what the current work adds to the literature. It is not sufficient to state that you are using longitudinal data when the prior studies also used longitudinal data (e.g., citation 16). Moreover, I found the rationale to focus solely on women lacking. In the Intro, the authors describe how prior work in this area did not stratify by sex. However, the authors here do not stratify by sex, rather, they just choose to look at women, without a solid justification for why.

In response, we added the following sentence as a justification of women only data in introduction.

“Sex-difference analysis in large epidemiology studies is commonly accepted analysis, which often leads to underestimation of importance of female sex in CVD risk due to under representation of women and male dominant CVD risk model [6]. The female subgroup analysis is warranted to elucidate female sex-specific CVD risk factors.”

[6]. Woodward M. Rationale and tutorial for analysing and reporting sex differences in cardiovascular associations. Heart. 2019;105(22):1701-1708. doi:10.1136/heartjnl-2019-315299

  1. For the life satisfaction measure, how did the authors deal with people who did not report having a spouse/partner or children?

Response: Participants without spouses and/or children were instructed to leave the items blank (skipped).  When there are missing data on items, imputed scores were calculated. For those with at least one response on items of spouses and children, scores of answered spouses/children items were averaged and average score is used for non response items. If none of spouses and children items were answered, then a  mean score of the rest of non spouse/children related items was used as a score.  The current reference [29] is the source of scoring.

Minor Comments

  1. The term “cardiovascular disease events” in the title is misleading – typically that is used to describe heart attacks and stroke. Would use the term “cardiovascular conditions” like it is described in the text.

Response: We have updated title to cardiovascular conditions.

  1. Citations are needed in line 32 and lines 45-48 of the Intro.

In response, we added citations.

Line 32:

  1. van Bussel, E.F.; Hoevenaar-Blom, M.P.; Poortvliet, R.K.E; Gussekloo, J.; van Dalen, J.W.; van Gool, W.A.; Richard, E.; Moll van Charante, E.P. Predictive value of traditional risk factors for cardiovascular disease in older people: A systematic review. Preventive Medicine. 2020, 132, 105986. doi:10.1016/j.ypmed.2020.105986

 Line 45-48

  1. Diener, E.; Suh, E.M.; Lucas, R.E.; Smith, H.L. Subjective Well-Being: Three Decades of Progress. Psychological Bulletin 1999, 125, 276–302, doi:10.1037/0033-2909.125.2.276.
  2. In Table 1, please report the breakdown of race/ethnicity rather than simply stating “% non-White”.

In response, we added break down in Table S1.

  1. The tables were difficult to read, particularly column 1, and the text spilling over onto multiple lines in Table 2.

This is an issue of a MDPI format and MS Word comparability. We will address your concern with a journal production team for readability once a paper is accepted.

  1. The Discussion would benefit from discussion of hypothesized mechanisms linking well-being and CVD conditions.

  In response, we have added the following to the discussion (p.8, lines 403-409).

“Potential mechanisms linking well-being [9] with CVD conditions include both behavioral and biological pathways. Greater well-being is thought to lead to more healthy behaviors and fewer unhealth behaviors thereby impacting one’s cardiovascular health. Well-being can strengthen restorative processes, such as physical activity and fruit/vegetable consumption, as well as reduce deteriorative processes, alcohol consumption [10]. Additionally, the protective influence of well-being on CVD health is also thought to occur via indirect biological pathways (i.e. inflammation, cortisol) [9].”

  1. Boehm JK, Kubzansky LD. The heart's content: the association between positive psychological well-being and cardiovascular health. Psychol Bull. 2012;138(4):655-691. doi:10.1037/a0027448
  2. Boehm JK. Positive psychological well-being and cardiovascular disease: Exploring mechanistic and developmental pathways. Soc Personal Psychol Compass. 2021;15(6):e12599. doi:10.1111/spc3.12599

Round 2

Reviewer 1 Report

I thank the authors for making efforts to address all my comments. The revision improves with respect to all issues I have raised in my review.

Author Response

Thank you for your constructive comments and feedback. 

Reviewer 2 Report

Overall, the authors adequately addressed the majority of my comments. Two minor issues remain:

1. The information regarding how spouse and children were handled for the Life Satisfaction measure was helpful - can this please be provided in the Methods section?

2. I appreciate the authors pointing out Table S4 for results excluding participants with pre-existing CVD. However, I did not see this noted anywhere in the text, and I think it is important to state more directly to reviewers that these analyses were conducted and where the results can be found, rather than just including this in the Supplementary Materials statements at the end of the manuscript.

Author Response

Thank you for your comments and input. We have addressed these by adding our responses to the main text with citations. All changes are yellow heighted in the manuscript.

Please find details of our responses below.

Overall, the authors adequately addressed the majority of my comments. Two minor issues remain:

  1. The information regarding how spouse and children were handled for the Life Satisfaction measure was helpful - can this please be provided in the Methods section?

In response, we added the following sentence in the methods section under life satisfaction scale description (yellow highlighted).

“ Participants without spouses and/or children were instructed to leave the items on spouses and children unanswered (blank). When there are missing data on these items, imputed scores were calculated. For those with at least one response on items of spouses and children, scores of answered spouses and children items were averaged and average score is used for non-response items. If none of spouses and children items were answered, then a mean score of the rest of non spouse/children related items was used as a score [29].”

  1. I appreciate the authors pointing out Table S4 for results excluding participants with pre-existing CVD. However, I did not see this noted anywhere in the text, and I think it is important to state more directly to reviewers that these analyses were conducted and where the results can be found, rather than just including this in the Supplementary Materials statements at the end of the manuscript.

In response, we added sensitivity analysis without pre-existing CVD condition in the method section, results, and discussion.

In methods section

“Furthermore, sensitivity analyses were conducted to test goodness of fit of excluding the pre-existing CVD condition from the models.”

In results section:

“Sensitivity analyses of excluding existing CVD condition from the models showed similar results of life satisfaction and SRH (Supplementary Materials Table S4). Without pre-existing CVD in the models, presence of diabetic condition was significant associated with increased CVD risk at 10 years and 19 years later (Table S4).”

In discussion section:

“Pre-existing CVD condition is a well established risk factor for the future CVD incidences and events among women, which is consistent with the current study models. Our sensitivity analyses of excluding pre-existing CVD condition from the prediction models on future CVD incidences showed diabetes as a significant CVD risk factor (Table S4), in contrast to our main finding of models with pre-existing CVD, presence of diabetes was not significantly associated with future CVD incidences (Table 3). This may be accounted for by the fact that diabetes is one of the known CVD risk factor [42,43] and is significantly correlated with pre-existing CVD condition (wave 1 χ2 = 40.56, p<.001; wave 2 χ2 = 80.74,  p<.001; wave 3 χ2 =72.37, p<.001). Including both pre-existing CVD and diabetes conditions may result in multicollinearity, the study used both AIC and C statistics and tested the model fit of inclusion of these covariates. Following model fit tests, the study’s current models with pre-existing CVD condition as baseline covariate showed better fit of models than without it.”

  1. Jeon‐Slaughter, H.; Chen, X.; Tsai, S.; Ramanan, B.; Ebrahimi, R. Developing an Internally Validated Veterans Affairs Women Cardiovascular Disease Risk Score Using Veterans Affairs National Electronic Health Records. Journal of the American Heart Association 2021, 10, e019217, doi:10.1161/JAHA.120.019217
  2. Goff, D.C.; Lloyd-Jones, D.M.; Bennett, G.; Coady, S.; D’Agostino, R.B.; Gibbons, R.; Greenland, P.; Lackland, D.T.; Levy, D.; O’Donnell, C.J.; et al. 2013 ACC/AHA Guideline on the Assessment of Cardiovascular Risk. Circulation 2014, 129, S49-S73, doi:10.1161/01.cir.0000437741.48606.98